# Prescribed-time consensus of multi-agent systems with distributed time-varying dynamic event-triggered strategy

Meilin Li
*School of Automation Engineering*
*University of Electronic Science and Technology of China*
Chengdu, China
meilinli0126@163.com

Tieshan Li
*School of Automation Engineering*
*University of Electronic Science and Technology of China*
Chengdu, China
tieshanli@126.com

*Abstract*—In this paper, the prescribed-time consensus control problem for multi-agent systems under undirected communication topology is considered. First, in order to save communication resource among agents, a dynamic event-triggered mechanism based on intermittent communication strategy is proposed. Additionally, the triggering conditions can be evaluated without real-time monitoring of neighboring agents' states and the communication resources of the whole closed-system can be greatly reduced. Then, a distributed control protocol based on a time-varying gain formulated by the parametric Lyapunov equation is presented to achieve the prescribed-time consensus control. Furthermore, the expression of the minimum inter-event time which has a strict positive lower bound is derived. Finally, the feasibility of the designed control method is validated through simulation results.

*Index Terms*—Multi-agent systems, prescribed-time control, consensus, dynamic event-triggered mechanism, intermittent communication.

## I. INTRODUCTION

In recent decades, due to its numerous applications, such as smart grids [1], maritime transportation [2], multi-robot operation [3], etc., the problem of coordinated control has attracted significant interest from researchers. Consensus control is a typical problem in coordinated control, where the goal is to design a control protocol so that all agents' states can converge to a uniform value. Currently, various consensus protocols have been designed to tackle this problem [4]–[7]. However, only asymptotic consensus can be achieved by these protocols, whereas in engineering practice, a faster convergence rate than exponential convergence is usually required. Therefore, how to solve this problem has garnered significant attention of many scholars.

Fortunately, prescribed-time control methods have been introduced to address the aforementioned issue. Since the convergence time of the prescribed-time control does not depend on the initial states and can be precisely specified by the user, the prescribed-time coordinated control problem of multi-agent systems (MASs) has attracted wide attention. In order to solve the prescribed-time consensus control problem of nonlinear MASs, distributed and fully distributed adaptive prescribed-time consensus control methods were proposed in [8], respectively. For the nonlinear MASs whose states are unmeasurable and affected by deception attacks, in [9], an observer was designed to estimate the unusable states, and a controller based on attack compensator was proposed to realize the prescribed-time consensus control. In [10], for the MASs with uncertain nonlinear terms, a prescribed-time consensus protocol based on adaptive neural network was proposed, which realized the consensus control in the prescribed-time. In [11], a distributed observer and a control protocol based on prescribed-time were proposed to realize the prescribed-time formation-containment control of MASs suffer from actuator faults. In [12], the output regulation of heterogeneous MASs under directed graph was studied. By designing the prescribed-time observer and the distributed prescribed-time control protocol, the prescribed-time output regulation control was finally realized.

It is important to note that the aforementioned control methods rely on continuous communication, which will increase the communication load of the controlled system. To address this issue, the event-triggered control methods were proposed [13]–[17]. In [18], the quantized consensus control of MASs with DOS attacks, bandwidth constraints and network delay was studied. By designing a quantized event-triggered secure consensus protocol, the consensus control was realized with less communication transmission and bit rate. To realise the consensus control of double integrator MASs with prescribed performance, an event-triggered framework and a self-triggered framework which can guarantee the prescribed performance specifications were proposed in [19]. For the sake of further prolong the average event-triggered time, dynamic event-triggered methods were proposed [20]–[22]. Subsequently, the researches of dynamic event-triggered control are extended to MASs. In [23], a dynamic event-triggered fuzzy controller was designed for the MASs with parameter uncertainties to realized the consensus control. The average

This work is supported in part by the National Natural Science Foundation of China under Grants 51939001, 62273072, 62203088, the Natural Science Foundation of Sichuan Province under Grant 2022NSFSC0903.

consensus control problem of MASs in the presence of false data injection attacks and privacy eavesdroppers was considered in [24]. By designing a dynamic event-triggered privacy protection method, consensus control was finally realized. In [25], the output regulation problem of nonlinear MASs under the condition of saving system resources and convergence in prescribed-time was considered. The control objectives were achieved by designing a novel dynamic event-triggered control method. In [26], the problem of prescribed-time control for first-order MASs was studied. The bipartite consensus was achieved within a prescribed-time by designing an event-triggered control method that does not require continuous communication.

Inspired by the above discussion, our goal is to address the prescribed-time consensus control problem for MASs under undirected communication topology. First, a novel dynamic time-varying event-triggered mechanism based on state estimators is designed. Through this way, the triggering conditions can be evaluated without continuously monitoring the states of neighboring agents and the communication resources of the whole closed-system can be greatly reduced. Then, a distributed prescribed-time consensus protocol based on a time-varying gain formulated by the parametric Lyapunov equation is designed. Finally, by the proposed dynamic event-triggered mechanism, we can derive the expression of the minimum inter-event time which has a strict positive lower bound.

## II. PROBLEM FORMULATION AND PRELIMINARIES

### A. Algebraic Graph Theory

Consider there is an undirected graph $G$. The Laplacian matrix of $G$ is $\mathcal{L} = \begin{cases} l_{ii} = \sum_{j=1}^{N} a_{ij}, i = j \\ l_{ij} = -a_{ij}, i \neq j \end{cases}$, where $a_{ij} = 1$ if agent $i$ can obtains information from agent $j$, otherwise $a_{ij} = 0$. Note that, the eigenvalues of the connected undirected graph $G$ are $0 = \lambda_1 < \ldots \leq \lambda_N$.

### B. Problem Formulation

Consider a MAS which consists of $N$ agents, and the dynamic of each agent is

$$\dot{x}_i(t) = Ax_i(t) + Bu_i(t), i = 1, ..., N, \quad (1)$$

where $x_i(t) \in R^m$ is the state vector; $u_i(t) \in R^n$ is the control input; $A \in R^{m \times m}$ and $B \in R^{m \times n}$ are system matrixes.

Note that, the following Assumptions and Lemma are necessary for realizing the consensus control.

**Assumption 1:** All the eigenvalues of matrix $A$ are on the imaginary axis.

**Assumption 2:** The undirected graph $G$ is connected.

**Lemma 1 [27]:** Let Assumption 1 hold, a parametric Lyapunov equation is shown as below:

$$A^{\mathrm{T}}P + PA - \sigma^{-1}PBB^{\mathrm{T}}P = -\gamma P, \quad (2)$$

where $\gamma > 0$, $P > 0$ and $0 < \sigma^{-1} \leq \lambda_2$.

Consider there is a Lyapunov equation

$$M(\gamma)(A + \frac{\gamma}{2}I_n)^{\mathrm{T}} + (A + \frac{\gamma}{2}I_n)M(\gamma) = \sigma^{-1}BB^{\mathrm{T}}, \quad (3)$$

where $M(\gamma) > 0$ and satisfying $P = M^{-1}(\gamma)$. In addition, one can obtain that $\frac{\alpha P}{n\gamma} \geq \frac{dP}{d\gamma} \geq \frac{P}{n\gamma} > 0$, $\alpha \geq 1$, $PBB^{\mathrm{T}}P \leq \sigma n\gamma P$, $\nu = (n\gamma + (\frac{1}{2}(n\gamma)^2 - \frac{1}{2}n\gamma^2 - tr(A^2))^{\frac{1}{2}})^2 - n\gamma^2 > 0$, $A^{\mathrm{T}}PA \leq \nu P$.

## III. MAIN RESULTS

In this section, a time-varying dynamic event-triggered approach is proposed for solving the prescribed-time consensus control of the MASs.

### A. Prescribed-time dynamic event-triggered control strategy design

First, we design a state estimator for agent $i$

$$\tilde{x}_i(t) = e^{A(t-t_l^i)}x_i(t_l^i), \forall t \in [t_l^i, t_{l+1}^i),$$

where $t_l^i$ is the $l$th event-triggered instant of agent $i$. The event-triggered instants $t_0^i$, $t_1^i$, ... will be determined by the triggering condition provided later.

Next, define a state threshold as

$$w_i(t) = \sum_{j=1}^{N} a_{ij}(\tilde{x}_i(t) - \tilde{x}_j(t)). \quad (4)$$

With (4), one has

$$w(t) = (L \otimes I_n)\tilde{x}(t). \quad (5)$$

Define a vector $\eta = [\eta_1^{\mathrm{T}}, ..., \eta_N^{\mathrm{T}}]^{\mathrm{T}}$, we have

$$\eta_i(t) = x_i(t) - \frac{1}{N}\sum_{j=1}^{N} x_j(t). \quad (6)$$

We can further get that

$$\eta(t) = (Q \otimes I_n)x(t), \quad (7)$$

where $Q = I_N - \frac{1}{N}\mathbf{1} \cdot \mathbf{1}^{\mathrm{T}}$ and $\mathbf{1} = [1, ..., 1]^{\mathrm{T}}$.

Define another vector $\tilde{\eta} = [\tilde{\eta}_1^{\mathrm{T}}, ..., \tilde{\eta}_N^{\mathrm{T}}]^{\mathrm{T}}$, we have

$$\tilde{\eta}_i(t) = \tilde{x}_i(t) - \frac{1}{N}\sum_{j=1}^{N} \tilde{x}_j(t), \quad (8)$$

and

$$\tilde{\eta}(t) = (Q \otimes I_n)\tilde{x}(t). \quad (9)$$

Then, define an error $e_i(t)$ as

$$e_i(t) = \tilde{x}_i(t) - x_i(t). \quad (10)$$

With (7) and (9), one has

$$\eta(t) = \tilde{\eta}(t) - (Q \otimes I_n)e(t). \quad (11)$$

Recalling (5) and (9), one has

$$w(t) = (L \otimes I_n)\tilde{\eta}(t), \quad (12)$$

where $QL = LQ = L$.

Then, we design

$$\gamma_0 = \frac{1}{\theta T}, \tag{13}$$

and

$$\gamma = \frac{T}{T-t}\gamma_0, \forall t \in [0, T), \tag{14}$$

where $0 < T < \infty$, $\gamma_0 > 0$, $\rho = \frac{\lambda_2}{2\lambda_N}$, $\theta = \frac{\rho n}{n+\alpha}$.

Design the distributed time-varying dynamic event-triggered mechanism as

$$t_{k+1}^i = \inf \left\{ t > t_k^i \middle| f_i(t) \leq 0 \right\}, \tag{15}$$

where

$$f_i(t) = \frac{\gamma\lambda_2}{2\lambda_N}\xi_i(t) + \frac{\gamma}{4\lambda_N}w_i^{\mathrm{T}}(t)Pw_i(t) \tag{16}$$
$$- \left(\frac{\gamma}{2\lambda_N} + \sigma n\gamma\right)\lambda_N^2 e_i^{\mathrm{T}}(t)Pe_i(t),$$

and

$$\dot{\xi}_i(t) = -\frac{\gamma\lambda_2}{2\lambda_N}\xi_i(t) + \frac{\gamma}{4\lambda_N}w_i^{\mathrm{T}}(t)Pw_i(t) \tag{17}$$
$$- \left(\frac{\gamma}{2\lambda_N} + \sigma n\gamma\right)\lambda_N^2 e_i^{\mathrm{T}}(t)Pe_i(t).$$

With (16) and (17), we can get that

$$\dot{\xi}_i(t) \geq -\frac{\gamma\lambda_2}{\lambda_N}\xi_i(t). \tag{18}$$

Then, we can further get that

$$\xi_i(t) \geq \mathrm{e}^{-\frac{\gamma\lambda_2}{\lambda_N}t}\xi_i(0) > 0, \forall t \in \left[t_l^i, t_{l+1}^i\right). \tag{19}$$

The prescribed-time consensus protocol is designed as

$$u_i(t) = Kw_i(t), \tag{20}$$

where $K = -B^{\mathrm{T}}P$.

Then, we can conclude the main results as shown below.

**Theorem 1:** Consider the MASs (1), let Assumption 1 and Assumption 2 hold, the prescribed-time consensus can be realised with the dynamic event-triggered intermittent communication mechanism (15) and the prescribed-time consensus protocol (20).

**Proof:** First, we choose a Lyapunov function

$$V(t) = \gamma \left( \eta^{\mathrm{T}}(t)(L \otimes P)\eta(t) + \sum_{i=1}^{N}\xi_i(t) \right). \tag{21}$$

Taking the time derivative of $V(t)$, one has

$$\dot{V}(t) = \dot{\gamma}\left(\eta^{\mathrm{T}}(t)(L \otimes P)\eta(t) + \sum_{i=1}^{N}\xi_i(t)\right)$$
$$+ 2\gamma\eta^{\mathrm{T}}(t)(L \otimes P)\dot{\eta}(t)$$
$$+ \frac{\alpha}{n}\dot{\gamma}\eta^{\mathrm{T}}(t)(L \otimes P)\eta(t)$$
$$+ \gamma\sum_{i=1}^{N}\dot{\xi}_i(t). \tag{22}$$

Recalling (1) and (20), one has

$$\dot{\eta}(t) = (Q \otimes I_n)\dot{x}(t)$$
$$= \left(I_N \otimes A - L \otimes BB^{\mathrm{T}}P\right)\eta(t)$$
$$- \left(L \otimes BB^{\mathrm{T}}P\right)e(t). \tag{23}$$

By substituting (23) into the term $2\eta^{\mathrm{T}}(t)(L \otimes P)\dot{\eta}(t)$, one has

$$2\eta^{\mathrm{T}}(t)(L \otimes P)\dot{\eta}(t)$$
$$\leq \eta^{\mathrm{T}}(t)\left(L \otimes \left(A^{\mathrm{T}}P + PA\right) - 2L^2 \otimes PBB^{\mathrm{T}}P\right)\eta(t)$$
$$- 2\eta^{\mathrm{T}}(t)\left(L^2 \otimes PBB^{\mathrm{T}}P\right)e(t)$$
$$\leq \eta^{\mathrm{T}}(t)\left(L \otimes \left(A^{\mathrm{T}}P + PA\right) - L^2 \otimes PBB^{\mathrm{T}}P\right)\eta(t)$$
$$+ e^{\mathrm{T}}(t)\left(L^2 \otimes PBB^{\mathrm{T}}P\right)e(t). \tag{24}$$

Define $\hat{\eta}(t) = \left(M^{\mathrm{T}} \otimes I_n\right)\eta(t) = \mathrm{col}\left(\hat{\eta}_1(t), ..., \hat{\eta}_N(t)\right)$ and $L = MUM^{\mathrm{T}}$, where $M$ is an orthogonal matrix which satisfies $M^{\mathrm{T}}M = I_N$ and $U = diag\{\lambda_1, ..., \lambda_N\}$. Then, recalling (24) we have

$$2\eta^{\mathrm{T}}(t)(L \otimes P)\dot{\eta}(t)$$
$$\leq \sum_{i=1}^{N}\lambda_i\hat{\eta}_i^{\mathrm{T}}(t)\left(A^{\mathrm{T}}P + PA - \lambda_i PBB^{\mathrm{T}}P\right)\hat{\eta}_i(t)$$
$$+ e^{\mathrm{T}}(t)\left(L^2 \otimes PBB^{\mathrm{T}}P\right)e(t). \tag{25}$$

Recalling (2), one has

$$A^{\mathrm{T}}P + PA - \lambda_2 PBB^{\mathrm{T}}P$$
$$\leq A^{\mathrm{T}}P + PA - \sigma^{-1}PBB^{\mathrm{T}}P$$
$$= -\gamma P. \tag{26}$$

Then, we can get that

$$2\eta^{\mathrm{T}}(t)(L \otimes P)\dot{\eta}(t)$$
$$\leq -\gamma\sum_{i=1}^{N}\lambda_i\hat{\eta}_i^{\mathrm{T}}(t)P\hat{\eta}_i(t)$$
$$+ e^{\mathrm{T}}(t)\left(L^2 \otimes PBB^{\mathrm{T}}P\right)e(t)$$
$$\leq -\frac{\gamma}{\lambda_N}\eta^{\mathrm{T}}(t)\left(L^2 \otimes P\right)\eta(t)$$
$$+ e^{\mathrm{T}}(t)\left(L^2 \otimes PBB^{\mathrm{T}}P\right)e(t). \tag{27}$$

By substituting (27) into (22), one has

$$\dot{V}(t) = \dot{\gamma}\left(\eta^{\mathrm{T}}(t)(L \otimes P)\eta(t) + \sum_{i=1}^{N}\xi_i(t)\right)$$
$$- \frac{\gamma^2}{\lambda_N}\eta^{\mathrm{T}}(t)\left(L^2 \otimes P\right)\eta(t)$$
$$+ \gamma e^{\mathrm{T}}(t)\left(L^2 \otimes PBB^{\mathrm{T}}P\right)e(t)$$
$$+ \frac{\alpha}{n}\dot{\gamma}\eta^{\mathrm{T}}(t)(L \otimes P)\eta(t)$$
$$+ \gamma\sum_{i=1}^{N}\dot{\xi}_i(t). \tag{28}$$

With (11) and Young's inequality, the term $-\frac{\gamma^2}{\lambda_N}\eta^{\mathrm{T}}(t)\left(L^2 \otimes P\right)\eta(t)$ can be rewritten as follows

$$-\frac{\gamma^2}{\lambda_N}\eta^{\mathrm{T}}(t)\left(L^2 \otimes P\right)\eta(t)$$
$$\leq -\frac{\gamma^2\lambda_2}{2\lambda_N}\eta^{\mathrm{T}}(t)\left(L \otimes P\right)\eta(t)$$
$$-\frac{\gamma^2}{2\lambda_N}\tilde{\eta}^{\mathrm{T}}(t)\left(L^2 \otimes P\right)\tilde{\eta}(t)$$
$$-\frac{\gamma^2}{2\lambda_N}e^{\mathrm{T}}(t)\left(L^2 \otimes P\right)e(t)$$
$$+\frac{\gamma^2}{\lambda_N}\tilde{\eta}^{\mathrm{T}}(t)\left(L^2 \otimes P\right)e(t)$$
$$\leq -\frac{\gamma^2\lambda_2}{2\lambda_N}\eta^{\mathrm{T}}(t)\left(L \otimes P\right)\eta(t)$$
$$-\frac{\gamma^2}{2\lambda_N}\tilde{\eta}^{\mathrm{T}}(t)\left(L^2 \otimes P\right)\tilde{\eta}(t)$$
$$-\frac{\gamma^2}{2\lambda_N}e^{\mathrm{T}}(t)\left(L^2 \otimes P\right)e(t)$$
$$+\frac{\gamma^2}{4\lambda_N}\tilde{\eta}^{\mathrm{T}}(t)\left(L^2 \otimes P\right)\tilde{\eta}(t)$$
$$+\frac{\gamma^2}{\lambda_N}e^{\mathrm{T}}(t)\left(L^2 \otimes P\right)e(t)$$
$$= -\frac{\gamma^2\lambda_2}{2\lambda_N}\eta^{\mathrm{T}}(t)\left(L \otimes P\right)\eta(t)$$
$$-\frac{\gamma^2}{4\lambda_N}\tilde{\eta}^{\mathrm{T}}(t)\left(L^2 \otimes P\right)\tilde{\eta}(t)$$
$$+\frac{\gamma^2}{2\lambda_N}e^{\mathrm{T}}(t)\left(L^2 \otimes P\right)e(t). \tag{29}$$

Then, one has

$$\dot{V}(t) = \dot{\gamma}\left(\eta^{\mathrm{T}}(t)\left(L \otimes P\right)\eta(t) + \sum_{i=1}^{N}\xi_i(t)\right)$$
$$-\frac{\gamma^2\lambda_2}{2\lambda_N}\eta^{\mathrm{T}}(t)\left(L \otimes P\right)\eta(t)$$
$$-\frac{\gamma^2}{4\lambda_N}\tilde{\eta}^{\mathrm{T}}(t)\left(L^2 \otimes P\right)\tilde{\eta}(t)$$
$$+\left(\frac{\gamma}{2\lambda_N} + \sigma n\gamma\right)\gamma e^{\mathrm{T}}(t)\left(L^2 \otimes P\right)e(t)$$
$$+\frac{\alpha}{n}\dot{\gamma}\eta^{\mathrm{T}}(t)\left(L \otimes P\right)\eta(t)$$
$$+\gamma\sum_{i=1}^{N}\dot{\xi}_i(t)$$
$$\leq \frac{n+\delta_c}{n\gamma}\dot{\gamma}V(t) - \frac{\gamma^2\lambda_2}{2\lambda_N}\eta^{\mathrm{T}}(t)\left(L \otimes P\right)\eta(t)$$
$$-\frac{\gamma^2}{4\lambda_N}w^{\mathrm{T}}(t)\left(I_N \otimes P\right)w(t)$$
$$+\left(\frac{\gamma}{2\lambda_N} + \sigma n\gamma\right)\gamma\lambda_N^2 e^{\mathrm{T}}(t)\left(I_N \otimes P\right)e(t)$$
$$+\gamma\sum_{i=1}^{N}\dot{\xi}_i(t). \tag{30}$$

Substituting (17) into (30), one has

$$\dot{V}(t) \leq \left(\frac{n+\alpha}{n\gamma}\dot{\gamma} - \gamma\rho\right)V(t)$$
$$= \left(\frac{(n+\alpha)\dot{\gamma} - n\gamma^2\rho}{n\gamma}\right)V(t)$$
$$= \left(\frac{(n+\alpha)\dot{\gamma} - \theta(n+\alpha)\gamma^2}{n\gamma}\right)V(t)$$
$$= \frac{(n+\alpha)\left(\dot{\gamma} - \theta\gamma^2\right)}{n\gamma}V(t). \tag{31}$$

Recalling (13) and (14), we can get that

$$\dot{\gamma} = \theta\gamma^2. \tag{32}$$

Then, we have

$$\dot{V}(t) \leq 0. \tag{33}$$

With (21) and (33), one has

$$\left\|\sum_{i=1}^{N}\sum_{j=1}^{N}a_{ij}\left(x_i(t) - x_j(t)\right)\right\|$$
$$\leq \sqrt{\frac{2\gamma(0)\left(\lambda_{\max}(L)\lambda_{\max}(P)\|x^{\mathrm{T}}(0)\|^2 + \sum_{i=1}^{N}\xi_i(0)\right)}{\gamma(t)\lambda_{\min}(P)}}, \tag{34}$$

where $\xi_i(0) = 0$.

With (34), we can easily get a conclusion that $\lim_{t \to T}\left\|\sum_{i=1}^{N}\sum_{j=1}^{N}a_{ij}\left(x_i(t) - x_j(t)\right)\right\| = 0$.

**Remark 1:** Recalling (14), we can observe that as $t$ approaches $T$, the $\gamma$ approaches infinity, which makes it impossible to implement in real world situations. Fortunately, this problem can be solved by redesign the parameter $\gamma$ as follows

$$\begin{cases} \gamma(t) = \frac{\bar{T}}{\bar{T}+\varepsilon-t}\gamma_0, \forall t \in \left[0, \bar{T}\right), \\ \bar{\gamma} = \gamma(t) = \frac{\bar{T}}{\varepsilon}\gamma_0, \forall t > \bar{T}, \end{cases} \tag{35}$$

where $\varepsilon$ is a very small positive constant. Reaclling (31) and (35), we have

$$\dot{V}(t) \leq \frac{-(n+\alpha)\theta\bar{\gamma}}{n}V(t). \tag{36}$$

Then, we can get that

$$\left\|\sum_{i=1}^{N}\sum_{j=1}^{N}a_{ij}\left(x_i(t) - x_j(t)\right)\right\|$$
$$\leq \sqrt{\frac{2V(\bar{T})\,\mathrm{e}^{-\frac{(n+\alpha)\theta\bar{\gamma}}{n}(t-\bar{T})}}{\bar{\gamma}(t)\lambda_{\min}(P)}}.$$

### B. Minimum inter-event time analysis

**Theorem 2:** With the dynamic event-triggered mechanasim (15), the minimum inter-event time has a strict positive lower bound

$$\tau_i = \int_0^{\frac{\kappa}{4\left(\frac{1}{2\lambda_N}+\sigma n\right)\lambda_N^3}}\frac{1}{\phi(\tau_i)}d\tau_i > 0, \tag{37}$$

where $\phi(\tau_i) = \sigma n\gamma^* + (2\upsilon(\gamma^*) + 2 + \sigma n\gamma^* + 2\rho\gamma^*)\tau_i + \left(\frac{\gamma^*}{2\lambda_N} + \sigma n\gamma^*\right)\lambda_N^2\tau_i^2$.

**Proof:** First, recalling the dynamic event-triggered function (16), one has that

$$f_i(t) \geq \frac{\gamma\kappa}{4\lambda_N}\left(\xi_i(t) + w_i^{\mathrm{T}}(t)Pw_i(t)\right)$$
$$- \left(\frac{\gamma}{2\lambda_N} + \sigma n\gamma\right)\lambda_N^2 e_i^{\mathrm{T}}(t)Pe_i(t), \quad (38)$$

where $\kappa = \min\{1, 2\lambda_2\}$.

Then, we can get that the inter-event times are constrained by the function

$$\Xi_i(t) = \frac{e_i^{\mathrm{T}}(t)Pe_i(t)}{\xi_i(t) + w_i^{\mathrm{T}}(t)Pw_i(t)}$$

going from 0 to $\frac{\kappa}{4\left(\frac{1}{2\lambda_N} + \sigma n\right)\lambda_N^3}$.

Taking time derivative of $\Xi_i(t)$ over interval $t \in [t_l^i, t_{l+1}^i)$, one obtains

$$\dot{\Xi}_i(t) = \frac{2e_i^{\mathrm{T}}(t)P\dot{e}_i(t) + \dot{\gamma}e_i^{\mathrm{T}}(t)\frac{dP}{d\gamma}e_i(t)}{\xi_i(t) + w_i^{\mathrm{T}}(t)Pw_i(t)}$$
$$- \Xi_i(t)\frac{\dot{\xi}_i(t) + 2w_i^{\mathrm{T}}(t)P\dot{w}_i(t) + \dot{\gamma}w_i^{\mathrm{T}}(t)\frac{dP}{d\gamma}w_i(t)}{\xi_i(t) + w_i^{\mathrm{T}}(t)Pw_i(t)}. \quad (39)$$

Recalling (4) and (10), one has

$$\dot{e}_i(t) = A\tilde{x}_i(t) - Ax_i(t) - Bu_i(t)$$
$$= Ae_i(t) + BB^{\mathrm{T}}Pw_i(t), \quad (40)$$

and

$$\dot{w}_i(t) = Aw_i(t). \quad (41)$$

With (40), we can further get that

$$2e_i^{\mathrm{T}}(t)P\dot{e}_i(t)$$
$$= 2e_i^{\mathrm{T}}(t)PAe_i(t) + 2e_i^{\mathrm{T}}(t)PBB^{\mathrm{T}}Pw_i(t)$$
$$\leq e_i^{\mathrm{T}}(t)A^{\mathrm{T}}PAe_i(t) + e_i^{\mathrm{T}}(t)Pe_i(t)$$
$$+ e_i^{\mathrm{T}}(t)PBB^{\mathrm{T}}Pe_i(t) + w_i^{\mathrm{T}}(t)PBB^{\mathrm{T}}Pw_i(t)$$
$$= (\upsilon + 1 + \sigma n\gamma)e_i^{\mathrm{T}}(t)Pe_i(t) + \sigma n\gamma w_i^{\mathrm{T}}(t)Pw_i(t), \quad (42)$$

and

$$\dot{\gamma}e_i^{\mathrm{T}}(t)\frac{dP}{d\gamma}e_i(t)$$
$$\leq \dot{\gamma}\frac{\alpha}{n\gamma}e_i^{\mathrm{T}}(t)Pe_i(t)$$
$$= \alpha_2\gamma\frac{\alpha}{n}e_i^{\mathrm{T}}(t)Pe_i(t)$$
$$\leq \rho\gamma e_i^{\mathrm{T}}(t)Pe_i(t). \quad (43)$$

With (41), one has that

$$- 2w_i^{\mathrm{T}}(t)P\dot{w}_i(t)$$
$$= - 2w_i^{\mathrm{T}}(t)PAw_i(t)$$
$$\leq w_i^{\mathrm{T}}(t)A^{\mathrm{T}}PAw_i(t) + w_i^{\mathrm{T}}(t)Pw_i(t)$$
$$= (\upsilon + 1)w_i^{\mathrm{T}}(t)Pw_i(t), \quad (44)$$

and

$$- \dot{\gamma}w_i^{\mathrm{T}}(t)\frac{dP}{d\gamma}w_i(t)$$
$$\leq - \frac{\dot{\gamma}}{n\gamma}w_i^{\mathrm{T}}(t)Pw_i(t)$$
$$= - \frac{\theta\gamma}{n}w_i^{\mathrm{T}}(t)Pw_i(t). \quad (45)$$

Substituting (42)-(45) into (39), we can get that

$$\dot{\Xi}_i(t) \leq \frac{(\upsilon + 1 + \sigma n\gamma + \rho\gamma)e_i^{\mathrm{T}}(t)Pe_i(t) + \sigma n\gamma w_i^{\mathrm{T}}(t)Pw_i(t)}{\xi_i(t) + w_i^{\mathrm{T}}(t)Pw_i(t)}$$
$$+ \Xi_i(t)\frac{\rho\gamma\xi_i(t) + (\upsilon + 1)w_i^{\mathrm{T}}(t)Pw_i(t)}{\xi_i(t) + w_i^{\mathrm{T}}(t)Pw_i(t)}$$
$$+ \Xi_i(t)\frac{\left(\frac{\gamma}{2\lambda_N} + \sigma n\gamma\right)\lambda_N^2 e_i^{\mathrm{T}}(t)Pe_i(t)}{\xi_i(t) + w_i^{\mathrm{T}}(t)Pw_i(t)}$$
$$\leq \sigma n\gamma + (2\upsilon + 2 + \sigma n\gamma + 2\rho\gamma)\Xi_i(t)$$
$$+ \left(\frac{\gamma}{2\lambda_N} + \sigma n\gamma\right)\lambda_N^2\Xi_i^2(t)$$
$$\leq \sigma n\gamma^* + (2\upsilon(\gamma^*) + 2 + \sigma n\gamma^* + 2\rho\gamma^*)\Xi_i(t)$$
$$+ \left(\frac{\gamma^*}{2\lambda_N} + \sigma n\gamma^*\right)\lambda_N^2\Xi_i^2(t). \quad (46)$$

Then, let's define

$$\dot{\Phi}_i(t) = \sigma n\gamma^* + (2\upsilon(\gamma^*) + 2 + \sigma n\gamma^* + 2\rho\gamma^*)\Phi_i(t)$$
$$+ \left(\frac{\gamma^*}{2\lambda_N} + \sigma n\gamma^*\right)\lambda_N^2\Phi_i^2(t), \quad (47)$$

where $\Xi_i(t_l^i) = \Phi_i(t_l^i) = 0$. Through (46) and (47), we can easily get that $\Xi_i(t) \leq \Phi_i(t)$.

Define $\Phi_i(t_l^i + \tau_i) = \frac{\kappa}{4\left(\frac{1}{2\lambda_N} + \sigma n\right)\lambda_N^3}$ and recalling (47), we can get that

$$\tau_i = t_l^i + \tau_i - t_l^i$$
$$= \int_0^{\frac{\kappa}{4\left(\frac{1}{2\lambda_N} + \sigma n\right)\lambda_N^3}} \frac{1}{\phi(\tau_i)}d\tau_i > 0. \quad (48)$$

As $\Phi_i(t)$ is a monotonically increasing function, it holds that $\Xi_i(t) \leq \Phi_i(t) \leq \frac{\kappa}{4\left(\frac{1}{2\lambda_N} + \sigma n\right)\lambda_N^3}, \forall t \in [t_l^i, t_l^i + \tau_i)$. Hence, we can conclude that no event is triggered when $t \in [t_l^i, t_l^i + \tau_i)$, and a strict positive lower bound can be obtained through (48).

## IV. SIMULATIONS

To verified the effectiveness of the time-varying dynamic event-triggered-based prescribed-time control strategy, the simulation results are given.

Consider there are 4 agents of the MAS and the communication topology is given in Fig. 1. Design the system matrices are $A = [0, 1; -2, 0]$ and $B = [0; 1]$. Some necessary initial values are defined as $x_1(0) = [-0.2; -0.3]$, $x_2(0) = [0.7; -0.5]$, $x_3(0) = [0.2; -0.8]$, $x_4(0) = [-0.3; 0.4]$, $\eta_1(0) = 3$, $\eta_2(0) = 3$, $\eta_3(0) = 3$, $\eta_4(0) = 3$. Choose the sampling interval as $0.0001s$ and $\bar{T}$ is chosen as $3s$.

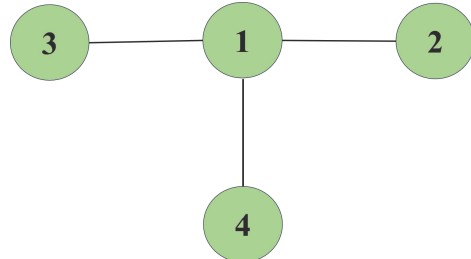

Fig. 1. The undirected graph of the MASs.

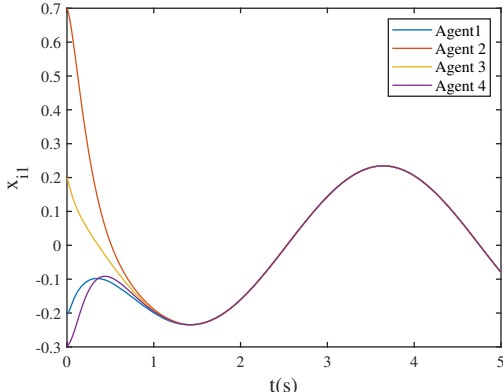

Fig. 2. The states of $x_{i1}$ of all the agents.

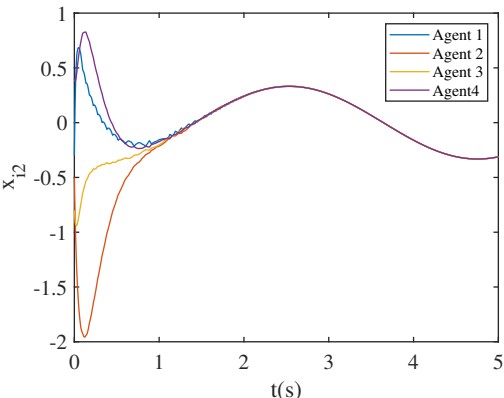

Fig. 3. The states of $x_{i2}$ of all the agents.

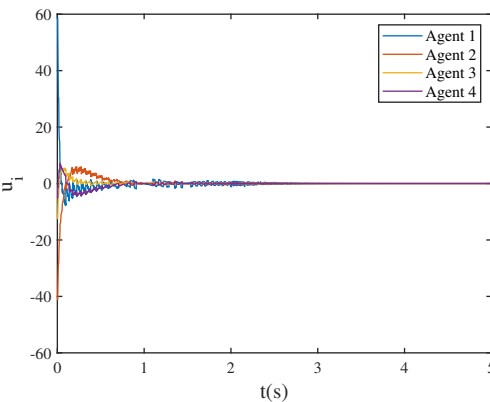

Fig. 4. The evolution of $u_i$ of all the agents.

From Figs. 2 and 3, it can be seen that the states of all the agents eventually convergence to a same value, indicating that the prescribed-time consensus has been achieved under the control method we proposed. Fig. 4 shows the evolution of $u_i$ of each agent. As illustrated in Fig. 5, the value of $\xi_i$ always remains greater than zero. To demonstrate the efficacy of the dynamic event-triggered method, Table I is given. From the event-triggered numbers, we can see that the communication transmissions among agents have been greatly reduced, which effectively conserving the communication resources of the MAS.

## V. CONCLUSION

In this paper, a prescribed-time consensus control method based on dynamic event-triggered intermittent communication strategy has been discussed. For the sake of reducing communication transmissions among agents and avoiding real-time monitoring of neighboring agents' states, a distributed time-varying dynamic event-triggered mechanism was proposed. Then, a distributed control protocol with time-varying control gain was given. Finally, the consensus control was realized with a prescribed-time. Additionally, it has been proven that

the Zeno phenomenon does not occur. In the future, we will focus on addressing the issue of prescribed-time consensus control under randomly switching topologies.

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

TABLE I
THE EVENT-TRIGGERED NUMBERS OF EACH AGENT

| Agent | 1 | 2 | 3 | 4 |
|---|---|---|---|---|
| Event-triggered numbers | 305 | 127 | 94 | 163 |

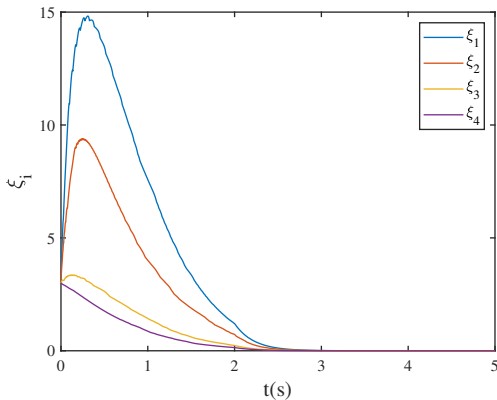

Fig. 5. The evolution of $xi_i$ of all the agents.

[7] M. Wang, J. Hu, A. Alsaedi and J. Cao, "Leader-Following Consensus Control of Unknown Nonlinear MASs Under False Data Injection Attacks," *IEEE Transactions on Network Science and Engineering*, doi: 10.1109/TNSE.2024.3433392.

[8] Z. u. A. Lodhi, K. Zhang, B. Zhou and H. Jiang, "Adaptive Prescribed-Time Consensus for a Class of Nonlinear Multi-Agent Networks by Bounded Time-Varying Protocols," *IEEE Transactions on Circuits and Systems I: Regular Papers*, doi: 10.1109/TCSI.2024.3430048.

[9] B. Niu, Y. Gao, G. Zhang, X. Zhao, H. Wang, D. Wang and C. Liu, "Adaptive Prescribed-Time Consensus Tracking Control Scheme of Nonlinear Multi-Agent Systems Under Deception Attacks," *IEEE Transactions on Automation Science and Engineering*, doi: 10.1109/TASE.2024.3408453.

[10] V. K. Singh, S. Kamal, S. Ghosh and T. N. Dinh, "Neuroadaptive Prescribed-Time Consensus of Uncertain Nonlinear Multi-Agent Systems," *IEEE Transactions on Circuits and Systems II: Express Briefs*, vol. 71, no. 1, pp. 296-300, 2024.

[11] X. Gong and X. Li, "Fault-Tolerant Practical Prescribed-Time Formation-Containment Control of Multi-Agent Systems on Directed Graphs," *IEEE Transactions on Network Science and Engineering*, vol. 11, no. 1, pp. 352-365, 2024.

[12] C. Chen, Y. Han, S. Zhu and Z. Zeng, "Prescribed-Time Cooperative Output Regulation of Heterogeneous Multiagent Systems," *IEEE Transactions on Industrial Informatics*, vol. 20, no. 2, pp. 2432-2443, 2024.

[13] P. Tabuada, "Event-Triggered Real-Time Scheduling of Stabilizing Control Tasks," *IEEE Transactions on Automatic Control*, vol. 52, no. 9, pp. 1680-1685, 2007.

[14] K. Zhang, B. Zhou, M. Hou and G. -R. Duan, "Practical Prescribed-Time Stabilization of a Class of Nonlinear Systems by Event-Triggered and Self-Triggered Control," *IEEE Transactions on Automatic Control*, vol. 69, no. 5, pp. 3426-3433, 2024.

[15] E. Garcia , Y. Cao and D. W. Casbeer, "Decentralized Event-Triggered Consensus with General Linear Dynamics," *Automatica*, vol. 50, no. 10, pp. 2633-2640, 2014.

[16] X. Guo, D. Zhang, J. Wang and C. K. Ahn, "Adaptive Memory Event-Triggered Observer-Based Control for Nonlinear Multi-Agent Systems Under DoS Attacks," *IEEE/CAA Journal of Automatica Sinica*, vol. 8, no. 10, pp. 1644-1656, 2021.

[17] A. Wang, X. Liao and H. He, "Event-Triggered Differentially Private Average Consensus for Multi-Agent Network," *IEEE/CAA Journal of Automatica Sinica*, vol. 6, no. 1, pp. 75-83, 2019.

[18] N. Lin and Q. Ling, "Event-triggered Quantized Consensus of Linear Multi-agent Systems under Asynchronous Denial-of-Service Attacks," *IEEE Transactions on Automatic Control*, doi: 10.1109/TAC.2024.3422230.

[19] Y. Hou, W. Hu, J. Li and T. Huang, "Prescribed Performance Control for Double-Integrator Multi-Agent Systems: A Unified Event-Triggered Consensus Framework," *IEEE Transactions on Circuits and Systems I: Regular Papers*, doi: 10.1109/TCSI.2024.3416400.

[20] A. Girard, "Dynamic Triggering Mechanisms for Event-Triggered Control," *IEEE Transactions on Automatic Control*, vol. 60, no. 7, pp. 1992-1997, 2015.

[21] X. Ge, Q. Han, L. Ding, Y. Wang and X. Zhang, "Dynamic Event-Triggered Distributed Coordination Control and its Applications: A Survey of Trends and Techniques," *IEEE Transactions on Systems, Man, and Cybernetics: Systems*, vol. 50, no. 9, pp. 3112-3125, 2020.

[22] K. Zhang, B. Zhou and G. Duan, "Event-Triggered and Self-Triggered Control of Discrete-Time Systems With Input Constraints," *IEEE Transactions on Systems, Man, and Cybernetics: Systems*, vol. 52, no. 3, pp. 1948-1957, 2022.

[23] Z. Du, X. Xie, Z. Qu, Y. Hu and V. Stojanovic, "Dynamic Event-Triggered Consensus Control for Interval Type-2 Fuzzy Multi-Agent Systems," *IEEE Transactions on Circuits and Systems I: Regular Papers*, vol. 71, no. 8, pp. 3857-3866, 2024.

[24] Y. Yang, J. Li, X. Wang, F. Ding, C. Dou and V. Kuzin, "Adaptive Dynamic Average Consensus Scheme With Preserving Privacy and Against False Data Injection Attacks: Dynamic Event-Triggered Mechanism," *IEEE Transactions on Vehicular Technology*, vol. 73, no. 6, pp. 7826-7837, 2024.

[25] Q. Hou and J. Dong, "Improved Event-Triggered Prescribed-Time Cooperative Output Regulation for Nonlinear Multiagent Systems," *IEEE Transactions on Circuits and Systems II: Express Briefs*, vol. 71, no. 4, pp. 2274-2278, 2024.

[26] X. Chen, H. Yu and F. Hao, "Prescribed-Time Event-Triggered Bipartite Consensus of Multiagent Systems," *IEEE Transactions on Cybernetics*, vol. 52, no. 4, pp. 2589-2598, 2022.

[27] B. Zhou, "Finite-Time Stabilization of Linear Systems by Bounded Linear Time-Varying Feedback," *Automatica*, vol. 113, pp. 108760, 2020.
