# OpenReview forum: "Prescribed-time consensus of multi-agent systems with distributed time-varying dynamic event-triggered strategy"
_IEEE.org/ICIST/2024/Conference — IEEE ICIST 2024 Conference Submission_

### Official Review · Reviewer_qmKP · 2024-08-22
**This article is very interesting and a good one**

**Rating:** 7
**Confidence:** 3

**Review:**

This paper proposed a prescribed-time consensus controller for multi-agent systems under undirected communication topology. The obtained result is valuable and can be accepted if the following problems can be clarified.
(1) In the introduction, the shortages of those relevant studies are suggested to be further summarized.
(2) What is the detailed process to obtain equation (28)? Please illustrate specifically.
(3) There exist several spelling and grammar errors. Please check carefully and further polish
(4) In the simulation section, more analysis can be added to better explain the main results of this paper, that's not enough.
(5) How to explain in detail that there is no Zeno behavior within the proposed control framework.

---

### Official Review · Reviewer_rm9w · 2024-08-23
**In this paper, the prescribed-time consensus control problem for multi-agent systems under undirected communication topology is considered. First, in order to save communication resource among agents, a dynamic event-triggered mechanism based on intermittent communication strategy is proposed. Additionally, the triggering conditions can be evaluated without real-time monitoring of neighboring agents’ states and the communication resources of the whole closed-system can be greatly reduced. Then, a distributed control protocol based on a time-varying gain formulated by the parametric Lyapunov equation is presented to achieve the prescribed-time consensus control. Furthermore, the expression of the minimum inter-event time which has a strict positive lower bound is derived. Finally, the feasibility of the designed control method is validated through simulation results.**

**Rating:** 7
**Confidence:** 3

**Review:**

(1)The structure of this paper is very compact in terms of content, but should each subsection begin with a brief introduction to the content as “III. MAIN RESULTS” does. The authors should revise as appropriate.
(2)The formula formatting in this paper is relatively standard, but try to see if you don't pay attention to the space issue, such as in formula (1) ‘ i = 1, ... , N,’ in ’...’ Are there spaces between them. The authors should double-check the whole text.
(3)The simulation diagrams in this paper are clear, but the edges of agents 1 and 2 in Figure 1 are incomplete. The authors should revise this carefully.

---

### Official Review · Reviewer_sYnS · 2024-08-26
**Prescribed-time consensus of multi-agent systems with distributed time-varying dynamic event-triggered strategy**

**Rating:** 7
**Confidence:** 2

**Review:**

In this paper, the prescribed-time consensus control problem for multi-agent systems under undirected communication topology is considered. The obtained result is valuable and can be accepted if the following problems can be clarified.
1. The paper should include comparisons against the existing literature to demonstrate its advantages.
2. In Section 3, the states of system had been defined. Why the vector \eta and \tilde \eta defined as (6)?
3. In the simulation, the paper should introduce the comparison of the state \x and the state estimator \tilde {x} of agent i.
4. The related references should be added to the Assumptions to show the rationality of them.

---

### Comment · Reviewer_qmKP · 2024-08-21
**This article is very interesting and a good one**

This paper proposed a prescribed-time consensus controller for multi-agent systems under undirected communication topology. The obtained result is valuable and can be accepted if the following problems can be clarified.
(1)	In the introduction, the shortages of those relevant studies are suggested to be further summarized.
(2)	What is the detailed process to obtain equation (28)? Please illustrate specifically.
(3)	There exist several spelling and grammar errors. Please check carefully and further polish
(4)	In the simulation section, more analysis can be added to better explain the main results of this paper, that's not enough.
(5)	How to explain in detail that there is no Zeno behavior within the proposed control framework.

---

### Decision · Program_Chairs · 2024-09-06

Accept (Oral)